# Efficiency of Human Resources in Public Hospitals: An Example from the Czech Republic

**DOI:** 10.3390/ijerph18094711

**Published:** 2021-04-28

**Authors:** Iveta Vrabková, Ivana Vaňková

**Affiliations:** Department of Public Economics, Faculty of Economics, VSB—Technical University of Ostrava, Sokolská třída 33, 702 00 Ostrava, Czech Republic; iveta.vrabkova@vsb.cz

**Keywords:** bootstrap, data envelopment analysis model, hospital efficiency, hospitals in public ownership, human resources in healthcare, performance

## Abstract

Healthcare is a highly sophisticated segment of the public sector, which requires not only highly professional and competent staff, but also a properly set ratio of healthcare professionals. In the Czech Republic, the state, as the main guarantor of health care, applied strong control through price and volume control. The aim of the paper is to define the differences in the technical efficiency of public hospitals, with regard to the size of hospitals and partial types of human resources. An input-oriented Data Envelopment Analysis model (DEA model) was chosen for modeling the technical efficiency of 47 public hospitals. The personnel performance concept of the evaluation of technical efficiency was further implemented in eight specific models, from the perspective of individual input variables relative to output variables and according to different assumptions regarding the character of economies of scale. The results of technical efficiency were analyzed using correlation, regression analysis, and the Bootstrap method. The least efficient hospitals in terms of hospital size are large hospitals, and the most balanced results have been achieved by medium-sized hospitals. The average efficiency rate in models that include all selected input and output variables is highest in medium-sized hospitals, with a value of 0.866 for CRS and an efficiency rate of 0.926 for VRS. The rationalization of human resources should be implemented in order not to reduce the quality of care provided.

## 1. Introduction

From an economic perspective, human resources in public health represent a highly specialized factor of production, irreplaceable in the short term. Public health and healthcare is an economically demanding sector where two thirds of financial resources in the individual organizations as well as the healthcare system are expended on labor force, and it is therefore essential to address the issues of the efficiency of human resources in order to ensure the sustainability and development of healthcare. The efficiency of human resources, especially health professionals, is a very topical, albeit controversial issue. Frequently mentioned, in this regard, is the disaccord between the efficiency, quality, and equality, but also the failure of the management of healthcare organizations and public administration, very often within the context of X-inefficiency [1,2]. Kozuń-Cieślak [3] state that the generally acknowledged positive relation between the human capital potential and prosperity cannot be verified because the investment into human capital in public health does not positively correlate with technical efficiency.

Human resources in public health are the executors of healthcare and the carriers of new and innovative therapeutic procedures, affecting the medical condition and quality of health of the population. Huanhuan et al. [4] state that enough individual categories of workers and the process of lifelong learning of health workers is reflected in the improving quality of health services. According to the OECD (Organisation for Economic Co-operation and Development) report [5], the Czech healthcare system works well and is improving. It is typical of the Czech public health system that it is under robust control by the government through price and volume regulation. Both the efficiency and the quality suffer from asymmetrical information shared between three types of actors—health insurance companies, healthcare providers (e.g., hospitals), and citizens. As a result, the system lacks proper motivational mechanisms and price signaling, a fact that indicates that there is much room for increasing the efficiency and quality.

Promoters and/or founders of public hospitals in the Czech Republic are the municipalities, regions, and the state. Public hospitals are mostly administered and/or established by the regions. Many regions, as hospital promoters, either implemented hospital privatization with a 100% ownership share of the region, or sold the hospitals to private owners. Some regions perform hospital mergers to streamline the management and increase the efficiency of the hospitals’ operation, creating large hospital consortia. Privatization and corporatization were also implemented in the East German hospital network, and the study of Lindlbauer et al. [6] shows that corporatization had a significantly more positive effect on the efficiency compared to the public hospitals of the original type and privatized hospitals. These approaches have their advocates and their opponents, and some authors argue that the steps taken by some of the regions intentionally obfuscate the boundaries between the private and public sectors in public health [7,8]. However, pressure on efficient healthcare production is applied very dynamically in the Czech conditions and abroad, whether in terms of the allocative efficiency (i.e., the efficiency of the use of financial resources within the individual healthcare segments) or in terms of technical efficiency (i.e., the ability of an organization to produce the maximum possible volume of outputs with the given volume of inputs and the given technologies) [9].

This research also reacts to the issues of the technical efficiency of hospitals, focusing on the specific human resources—physicians, nurses, and other staff—as inputs, in relation to the main hospital production outputs attained, i.e., the number of “attended” patients in bed, in outpatient wards, and in operating rooms.

The article aims to outline the differences in the technical efficiency of hospitals administered or established by the public administration, with respect to the subtypes of human resources and the hospital sizes.

Two research questions, RQ1 and RQ2, were formed to support the objective.


*RQ1: Which human resource, with respect to the outputs attained, affects the degree of a hospital’s inefficiency the most?*


The first research question is based on the assumption that the least efficient human resource of a hospital are other workers, followed by the general nurses, while inefficiency is affected the least by the physicians.


*RQ2: Is it true that big hospitals are more efficient than small and medium-sized hospitals?*


It is generally presumed that larger organizations are able to attain so-called returns to scale. However, consideration might also be given to the manifestations of the X-inefficiency phenomenon, linked with the monopolistic behavior of market operators and public administration and service organizations. The general basis of the monopoly position of the public service providers (hospitals) is their original, non-transferrable legal authorization for the provision of services and management of public funds and assets. It is therefore possible that, in practice, hospital consortia and big university hospitals manifest themselves as monopolies or oligopolies.

The article has five parts. The first part is this introduction; part two focuses on synthesizing the knowledge of the evaluation of the hospitals’ technical efficiency and other healthcare according to the Data Envelopment Analysis Model (hereinafter the DEA Model); the third part deals with the research methodology, including the objective and research questions; part four presents the results of the calculations and analyses; and the fifth part contains the conclusion and discussion about the problem in relation to the results attained.

## 2. Materials and Methods

### 2.1. Literature Review: Evaluation of Technical Efficiency According to the DEA Model

The healthcare service performance evaluation has become a fundamental basis for decision-making on the level of healthcare facility management as well as other healthcare policy actors, relating to strategic issues within a hospital’s operation. The evaluation of health services has been addressed by a number of authors worldwide, a fact documented by a multitude of articles published on this subject. Multi-criteria methods are widely used for the evaluation as tools that are able to assess the efficiency of production units and show the opportunities for the improvement of inefficient units, but also to identify exemplary units. One of the tools able to determine the rate of technical efficiency of production units is the Data Envelopment Analysis Model. As shown in the articles specified below, an evaluation of technical efficiency over a multiannual period points to a greater stability of hospitals and thus to more relevant analysis results.

Hospital human resources are always modelled as inputs, while physicians and nurses are most frequently mentioned by the authors. But in some cases, they include all hospital employees among the human resources. Quite frequently, they combine human resources with other technical (e.g., number of beds, instrumentation) or financial (organizational) inputs in absolute or relative terms.

Pirani et al. [10] focused on the evaluation of the efficiency of public hospitals between 2012 and 2016. Their pivotal method was the output-oriented DEA Model positing variable returns to scale, while the input variables comprised the number of hospital admissions, the number of nurses, and the number of available beds. The output variables included the average length of stay and the bed turnover interval. The results show that technical efficiency was lower than scale efficiency, and the regression analysis confirmed a statistically important relationship between the variables of the average length of stay and the number of available beds.

Likewise, Ghahremanloo et al. [11] point to the importance of performance evaluation as a relevant tool for hospital management. They attempted to develop the traditional DEA Model with new elements that would, in addition to identifying efficient and inefficient units, allow for the introduction of options for changing the input and output indicators, so that the units would become efficient. The proposed new DEA Model includes the evaluation of the overall hospital efficiency. The model’s input indicators comprise the number of healthcare professionals, the number of other staff, and the number of beds. The output indicators include the bed occupancy rate and the bed turnover rate. The authors believe the efficiency- and efficacy-based performance evaluation may aid the managers in assessing whether the organization meets its targets, thus facilitating the decision-making process in the area of human resources as well as safeguarding the hospital’s production.

The international evaluation of hospitals was addressed by Varabyova et al. [12]. Their research was focused on applying the advanced non-parametric methods (such as DEA and FDH) to evaluate certain Italian and German hospitals. Once again, their input indicators were the number of beds, the number of physicians and the number of nurses (the personnel data are specified in a full-time equivalent), while the output indicators were the selected inpatient adjusted and day cases. The hospital performance evaluation results were interpreted according to the number of beds and the hospital’s ownership structure (public/private). It was concluded that the main differences in the hospital degree of efficiency were found in hospitals which differed in the range of medical branches offered, i.e., in their specialization.

Kocisova, K. et al. [13] used the Data Envelopment Analysis Model to compare the degree of efficiency of Polish hospitals in the meso perspective. This means that indicators selected for the evaluation of hospital efficiency were aggregated on the regional level (Polish provinces). The results were interpreted on the individual provinces. It was found that 11 of the sixteen provinces were inefficient. The authors then specified an efficient province, one that became a specimen for the inefficient units. They also examined samples of hospitals aggregated in the given province.

An evaluation of 22 hospitals in Spain was performed by Caballer-Tarazona [14]. The authors state that the evaluation of hospital efficiency is of the utmost importance because large amounts of public resources flow into the public health system which should be spent efficiently. The evaluation was, again, performed using the DEA Method with defined inputs (number of physicians, number of beds) and outputs (number of consultations, number of treatments, number of surgeries).

Khushalani, Yasar [15] inquired into the efficiency of production in hospitals between 2009 and 2013 using the Dynamic Data Envelopment Analysis with regard to the nature of the individual hospitals (healthcare comprehensiveness). The healthcare quality factor was incorporated in the modeling. Using the Pearson’s correlation coefficient, the authors confirmed the relationship between the efficiency of quality production and the efficiency of health and surgical care production. They concluded that municipal and university hospitals were less likely to increase their efficiency of quality production.

A study published by Kohl et al. [16] reviewed 262 articles that applied the DEA Method in public health, with a special focus on the hospital sector. Research publications were classified in clusters according to the purpose of analysis—pure DEA efficiency analysis, developments or applications of new methodologies, specific management questions, i.e., analyzing the effects of managerial specification, such as ownership, on hospital efficiency and a survey on the effects of reforms, i.e., researching the impact of policy making, such as reforms of health systems, on hospital efficiency. In conclusion, the authors assessed possible the pitfalls when performing analyses using the DEA method, and specified the recommendations for the application of the DEA method as a tool which the architects of health policies as well as the hospital managers should use when making decisions on ensuring the hospital’s production.

### 2.2. Research Methodology

The modeling of the technical efficiency of hospital human resources was implemented according to the DEA Model, distinguishing between the constant and variable returns to scale. The evaluation of hospital efficiency was performed by transforming the inputs to outputs in relation to other units of the given set [17]. The outputs had a maximization nature, while the inputs had a minimization nature. The first DEA Model was formulated in the study published by Charnes, Cooper, Rhodes [18]. This model is based on the assumption of constant returns to scale and maximizes the efficiency of the evaluated production unit under the condition that the efficiency of all other units is less than or equal to one. Models with variable returns to scale were first published in an article of Barker et al. [18]. The prerequisite of variable returns to scale is the fact that an increase or decrease in the inputs does not lead to a proportional increase or decrease in the outputs. In the case of variable returns to scale, the efficiency of the production unit is higher (or, more precisely, is not lower) than the efficiency in the case of constant returns to scale [18,19,20].

The modeling of technical efficiency was performed using the input-oriented model that expects that inefficient units should reduce their inputs with respect to the outputs attained. However, it is also a well-known fact that reductions in key human resources (physicians, general nurses, and midwives) has a negative impact on the quality of the services provided in both public health and social services [21]. Therefore, inefficient hospitals will not be advised to reduce the number of physicians and general nurses. The results attained will be used to compare the efficiency of a hospital’s personnel structure taking into account the size of the hospitals, which, as specified above, are administratively merged, thus creating abnormally large entities (consortia).

The description of the production units evaluated—public hospitals, the inputs and outputs, evaluation models, and the basic outline of the input-oriented DEA Model with constant returns to scale and variable returns to scale—is specified below.

The attained results of the calculation of technical efficiency according to the DEA (using the DEA Frontier Add-In for Microsoft Excel (Microsoft Corp, Redmont, WA, USA)) were analyzed using standard statistical methods, especially correlation (Pearson’s correlation coefficient) and regression (simple regression). Moreover, the Bootstrap was used, similarly to other studies [18,22]. Statistical analyses were performed using the IBM SPSS software (IBM, Armonk, NY, USA). Bootstrapping is a computer technique that understands a sample as a basic file, from which the computer generates new sample files (so-called Bootstrap samples), having the same size as the original file. From the above procedure, the standard deviation can be estimated, which is used to calculate the confidence interval [23].

### 2.3. Production Units Evaluated in the Context of the Health System

As of 31 December 2019, a total of 194 public and private hospitals operated in the CR, with the total capacity of 60,633 beds. A hospital fulfils a number of functions. Its basic function is providing diagnostic and therapeutic services and other supporting activities which predominantly fall within secondary and tertiary healthcare. From the functional viewpoint, hospitals may be subdivided into several categories. Primarily, hospitals may be subdivided, according to the health services they provide, into university hospitals, hospitals that provide acute inpatient care, and aftercare hospitals focusing on aftercare and long-term care. In terms of acute inpatient care valuation, it is suitable to categorize hospitals according to their size (the number of beds) and comprehensiveness of care, and more specifically into the following groups: university hospitals and big regional hospitals; healthcare facilities providing highly specialized care; regional and provincial hospitals providing comprehensive care; and provincial hospitals with lower comprehensiveness of care. Other commonly used typologies of hospitals within research areas include the categorization according to the average treatment period; promoters/founders (ministry, region, municipality, natural person, church organization, other legal entity); type of ownership of healthcare facilities; or the ratio between the number of specialized healthcare professionals and other employees.

Hospitals provide acute inpatient and outpatient care and belong to the pillars of healthcare, together with specialized outpatient clinics and health centers. According to the Czech Statistical Office [24], 40.25% of all expenses for public health were spent between 2010 and 2018 in the Czech Republic for inpatient and outpatient care. The hospital network comprises both public and private hospitals, while public hospitals are unambiguously dominant. It must be added in this regard that after 1989, the Czech public health system was hugely affected by political, economic, and social changes, strongly accentuating liberalization, deconcentration, and decentralization. Significant changes implemented in the last 30 years were aimed at making the public health system more efficient, whether from the perspective of the operation of the hospitals or in terms of changes in the hospital care funding. The transformation was related to the changes in hospital promoters, as well as to the changes in the owners of hospital assets. This brought a large difference of opinions, not just in the political arena, but also with the public, who feared problems with the real availability of the hospital by car [7]. In addition, Nuti et al. [25] point out that many of the changes that are being made in the healthcare segment are based on both external factors and the development of the organizational framework for healthcare.

This research focuses on public hospitals that provide comprehensive acute inpatient care. This means that the individual hospitals provide inpatient care in at least three of seven basic medical specialties (internal medicine; surgery; pediatrics; gynecology and obstetrics; anesthesiology and intensive care medicine; neonatology; neurology). Public hospitals are those administered (contributory organizations) or established (public limited companies) by public authorities.

The set of 47 public hospitals investigated was assessed as a whole. The efficiency results were also reviewed from the hospital size perspective. The hospital size criterion chosen was the number of beds, with the resulting three hospital sizes:26 hospitals with a number of beds <499—indicated as Small (S);13 hospitals with a number of beds >500 and <999—indicated as Medium (M);8 hospitals with a number of beds >1000—indicated as Big (B).

### 2.4. Inputs and Outputs

Inputs and outputs were evaluated, the values of which corresponded to the average value for three years, 2017–2019. The three-year average was chosen intentionally, for two reasons. The first reason was the fact that results for one year may be misleading in some hospitals (e.g., due to investments or extraordinary events); the second was the need to cover the most current staffing and performance condition of the hospitals before the pandemic (COVID-19), i.e., the year 2019.

Three input indicators were chosen to fulfil the objective of the article—the number of physicians, the number of nurses, and the number of other staff. An optimal ratio between the healthcare professionals and non-medical staff is crucial for the efficient operation of a hospital. It must be noted that this ratio cannot be the same in each healthcare facility because it mainly depends on the proportional design of the hospital and whether the hospital itself ensures all necessary activities or outsources certain services. In the Czech Republic, hospital staffing is specified in two laws: in Act No. 95/2004 Coll., on the requirements for the acquisition of recognition of professional competence to practice the profession of physician, dentist, and pharmacist, as amended [26]; and Act No. 96/2004 Coll., on the conditions for attaining and recognizing qualifications to perform professions other than medical professions and to perform activities relating to health care provision and on the amendment to some other acts, as amended [27]. The data structure is based on the requirements for human resources in the individual healthcare wards in accordance with Decree No. 99/2012 Coll., on the minimum staffing requirements for health services [28]. Indicators regarding the number of employees are based on the converted average annual number of registered employees and contractual workers.

The output indicators included the number of outpatient treatments, the number of hospitalized patients, and the number of surgeries. For the purpose of the analysis, data were taken from the annual reports of the individual hospitals for the periods in view, as well as from the National Register of Hospitalized Patients, the National Register of Healthcare Providers, and the National Register of Healthcare Professionals. The data were always valid as of the last day of the given calendar year. The indicators were chosen so that the condition of data availability and relevance was fulfilled in order to ensure clear identification, and so that the sample size and data quality did not significantly decrease the informative value and data comparison. As specified above, data for three years were investigated, in particular to minimize the influence of random, unrepeated fluctuation in the hospital production.

Table 1 below specifies the input and output indicators, including the detailed characteristics.

Table 2, below, characterizes the statistics of input and output indicators. The table makes it clear that the set of 47 public hospitals comprises groups of hospitals with substantially different sizes (according to the number of beds), which affects the defined indicators. The healthcare staffing was determined according to the expertise of the individual healthcare professionals, as well as the kind and specialty of healthcare. Within acute inpatient care, the workload of medical staff was established according to the number of beds in the given ward (see Act No. 99/2012 Coll.). The table shows that the largest values are attained by state-owned university hospitals—significant backbone hospitals with highly specialized healthcare and the most comprehensive provision of health services. Six out of 10 university hospitals were included in the set. On the contrary, the lowest values were attained by provincial hospitals which comply with the methodical criteria for entry in the hospital sample and perform surgeries within the individual specialties.

### 2.5. Models

The staff-performance estimation of efficiency according to the DEA Model is implemented using eight specific models. The first two models contain all three inputs and outputs and estimate the efficiency with constant returns to scale (CRS) and the efficiency with variable returns to scale (VRS). These models are indicated as total (T). Consequently, the partial efficiency is estimated from the perspective of the individual inputs (x1, x2, x3) compared to all outputs, in terms of both the CRS and VRS—see Table 3.

### 2.6. Data Envelopment Analysis Model: Input-Oriented Models

The Data Envelopment Analysis Model is a non-parametric method widely used for the evaluation of relative efficiency and performance of a set of decision-making units (DMUs). It is based on the maximization of the weighted sum of outputs produced by the unit evaluated divided by the weighted sum of inputs of the same unit, while assuming this ratio must be less than or equal to 1 for all other units. Unit homogeneity is an important prerequisite for the use of the DEA Models.

The DEA Model allows for the discrimination between input and output orientation during the calculation. This article only uses input-oriented models. Furthermore, the DEA Model discriminates the returns to scale, especially the constant returns to scale (CRS), which refer to the overall efficiency, and the variable returns that estimate the net efficiency. The breakdown of technical efficiency, calculated as the quotient of CRS and VRS, also allows for the determination of the scale’s efficiency [17,18,20,29].

The following is a mathematical formulation of the input-oriented DEA Model under the conditions of constant returns to scale:
(1)$$\begin{array}{ccc} \mathrm{maximize}\;\mathrm{eff}\left(U_{q}\right)= & \frac{\sum_{k}^{r}{\;u}_{k}y_{\mathrm{kq}\;}}{\sum_{i}^{m}v_{i}x_{\mathrm{iq}}}, & \\ \mathrm{under}\;\mathrm{the}\;\mathrm{conditions}\; & \frac{\sum_{k}^{r}{\;u}_{k}y_{\mathrm{kj}}}{\sum_{i}^{m}v_{i}x_{\mathrm{jk}}}\;\leq \;1,\; & \;j=1,\;2,\;\ldots ,\;n,\\ & u_{k}\;\geq \;\varepsilon ,\; & \;k=1,\;2,\ldots ,\;r,\\ & v_{i}\;\geq \;\varepsilon ,\;\; & i=1,\;2,\ldots ,\;m, \end{array}$$
where the unit efficiency U_q_ is expressed by eff(U_q_), ε is a constant ensuing the condition of positiveness of weights of the inputs and outputs, x_ij_, i = 1, 2, …, m, j = 1, 2, …, n indicates the value of the i-th input for the unit U_j_, and y_kj_, k = 1, 2, …, r, j = 1, 2, …, n indicates the value of the i-th input for the unit U_j_.

The calculation of the input-oriented DEA Model with variable returns to scale may be expressed as follows:
(2)$$\begin{array}{cccc} \mathrm{maximize}\;Ѳ\left(U_{q}\right)= & \sum_{k}^{r}{\;u}_{k\;}y_{\mathrm{kq}}+µ, & \\ \mathrm{under}\;\mathrm{the}\;\mathrm{conditions}\; & \sum_{k}^{r}{\;u}_{k}{\;y}_{\mathrm{kj}}+µ\;\leq \sum_{i}^{m}{\;v}_{i}{\;x}_{\mathrm{ij}}, & \;j=1,\;2,\;\ldots ,\;n,\\ & \sum_{i}^{m}{\;v}_{i}{\;x}_{\mathrm{iq}}=1, & \\ & u_{k}\;\geq \;\varepsilon ,\;\;\; & i=1,\;2,\;\ldots ,\;r,\\ & v_{i}\;\geq \;\varepsilon ,\;\;\; & & j=1,\;2,\;\ldots ,\;m,\\ & µ–\;\mathrm{free}. \end{array}$$

The interpretation of results for the individual units is analogous to the CCR Model—the efficient unit θ${(U}_{q})$ equals 1, while θ${(U}_{q})\;$< 1 applies to efficient non-efficient units [30].

## 3. Results

### 3.1. Total and Partial Models

The results of the hospital efficiency calculations according to the input-oriented CRS and VRS models show that hospitals (DMUs) are less efficient in the CRS models, as expected, compared to the VRS models. However, the differences between the models that discriminate between the returns to scale are not large, averaging 8%. The above is confirmed by the number of fully efficient (e = 1, 100%) hospitals; the T_CRS model comprises seven fully efficient hospitals and the T_VRS model has 18 fully efficient hospitals. It is further true that if a hospital is fully efficient in the CRS models, it is also fully efficient in the VRS models.

The average results of partial models focusing on the individual inputs (x1—physicians, x2—nurses, and x3—other staff) in relation to all results (y1—number of hospitalized patients, y2—number of outpatient examinations, y3—number of surgeries) show that x1 (physicians) is the least efficient input in terms of both the constant and variable returns to scale. Input x3 (other staff) shows the second-lowest average results. Input x2 (nurses) attains the best results on average. Inefficient hospitals, with the efficiency level below the average efficiency (CRS 83% and VRS 91%) should seriously consider reducing some of the inputs or increasing some procedures—outputs.

Table 4 shows the average values in the models of the overall technical efficiency CRS and net technical efficiency VRS including the 95% confidence interval, which expresses the upper and lower limits of the average values of efficiency according to the Bootstrap.

The confidence interval was also used to specify the inefficiency rate (IR), as follows:mild inefficiency: 1> IR ≥ Upper;moderate inefficiency: Upper > IR ≥ Lower;strong inefficiency: IR < Lower.

The distribution of results between the individual inefficiency levels (mild, moderate, strong) in the individual models is affected by the orientation of the returns to scale model, see Figure 1. In the CRS models, inefficiency is divided into the individual levels more proportionately than in the VRS models. The results may also suggest that the VRS models are more likely to tend to the extremes within the result distribution. The partial models confirm that models x1_CRS/VRS attain the worst results, while the best results are attained by x2_CRS/VRS.

The results of the correlation analysis using the Pearson’s correlation coefficient are shown in Table 5. The correlation coefficients monitor whether the increase or decrease of a value of one attribute causes the increase or decrease of another attribute. The correlation analysis attributes are the results of the total as well as partial models of efficiency calculations. The correlation matrix shows that there is a positive dependence between the individual attributes (an increase of one attribute leads to an increase of another attribute), which has been verified on the level of significance α = 0.001. Logically, the strongest correlation is between the attributes outside the individual models, CRS and VRS. However, correlation was also documented between the attributes across the CRS and VRS models. The lowest intensity of reaction to the growth of the value of the total efficiency models (T_CRS and T_VRS) was detected in the partial models x1_CRS and x1_VRS, i.e., those oriented to the evaluation of efficiency of the physicians. On the contrary, the strongest reaction to the growth of the efficiency value was reported in the partial models x3_CRS and x3_VRS at T_CRS and partial models x2_CRS and x2_VRS at T_VRS.

### 3.2. Models According to Hospital Size

The set of 47 hospitals in view was divided into three groups with different sizes (26 Small, 13 Medium, 8 Big), to discriminate the efficiency results with respect to hospital size.

The subdivision of the efficiency calculation results in terms of hospital size confirms that the highest number of fully efficient hospitals belongs to the category of small hospitals; on the contrary, the lowest number of fully efficient hospitals belongs to the category of big hospitals. Big hospitals, especially university hospitals, are the least efficient within the set in view. Only three big hospitals attain full efficiency in the VRS models, the best of them being H29, which is fully efficient in both the CRS and VRS models. The most balanced are attained by medium-sized hospitals—see Table 6.

Partial models are represented graphically using the regression models (T_CRS and T_VRS is the dependent variable), confirming the results specified in Table 6, including the fact that the worst results are attained in the partial model focused on x1, i.e., physicians. The reason for determining the dependent variables in the form of T_CRS and T_VRS is that the partial models can be used to explain and predict the value of the dependent variable, not the other way around.

The charts shown below (Figure 2a–f) make it clear that medium-sized hospitals attain the most balanced results. Big hospitals are the least efficient group of hospitals. Small hospitals attained the best as well as the worst results. Comparison of the CRS and VRS models (e.g., a and b) shows that in the CRS models, the results tend to accumulate toward the average in a larger extent compared to the VRS models.

Regression models also show that the partial models x2_CRS, x2_VRS have the most significant positive effect on the growth of the value of the overall models. The x1_CRS, x3_CRS, x1_VRS and x3_VRS models have a comparable effect on the dependent variables T_CRS and T_VRS.

## 4. Discussion

The analysis of a sample of 47 public hospitals made it clear that the differences in the technical efficiency of public hospitals were caused by the varying, but not very dissimilar efficiency of the individual type of human resources (inputs)—physicians, nurses, and other staff. The results of technical efficiency were affected by the hospital size (the number of beds) as well as the returns to scale.

The least efficient human resource in relation to the outputs in view (the number of hospitalized patients, the number of outpatient examinations, and the number of surgeries) is the input physicians, as the average efficiency of this inputs is 8–12% lower than the total average technical efficiency values, which range between 91% and 47%. The most numerous human resource in the hospitals is other staff; the average efficiency of this input is comparable to, or no more than 9% lower than, the total average technical efficiency values. The third human resource—nurses—is the least inefficient human resource, as its average technical efficiency value is no more than 4% lower than the total average technical efficiency values. Likewise, a finer resolution of the results according to the inefficiency level (mild, moderate, strong) confirms the above findings, although a varying distribution of inefficiency in the CRS and VRS models may be found. These results support the opinion that the CRS models are more accurate in this estimate of technical efficiency, when limiting the chosen inputs and outputs and the specific set of hospital DMUs, and they are better in revealing inefficiency in both the total and partial representation.

The above results may indicate—within the context of the evaluation logic according to the input-oriented DEA Model—that inefficient hospitals should primarily consider reducing the number of physicians and other medical staff. However, this recommendation must be considered individually in the conditions of particular hospitals, especially those showing an extensive inefficiency degree, even if this includes at least 31% of the hospitals in view.

The results also suggest that the initial hypothesis—that the least efficient input is other staff—was not correct.

Inefficiency in the hospitals in view depends on their size, as documented by the technical efficiency results when divided into three size groups in the hospitals in view. Big hospitals (university hospitals and consortia), although least represented in the set in view (eight hospitals), are the least efficient, and their results lie under the average of the technical efficiency value of the whole set, by at least 15%. Small and medium-sized hospitals show comparable results within the evaluation, while medium-sized hospitals are slightly better. In terms of hospital size and specific input, the results are identical to the above findings, as shown through regression analysis, proving that the type of inputs affects the overall technical efficiency results.

The evaluation of the technical efficiency results according to hospital size shows that big hospitals are not the leaders within the set in view—on the contrary, they are the least efficient units of the set in view. The initial hypothesis about the returns to scale is not true if human resources in public health are the factor of savings. In this regard, the connection between the hospital size and the subsequent, almost monopolistic position in the region is the decisive factor, and the basis for the occurrence of X-inefficiency.

The evaluation of technical efficiency and the results must also be perceived in a wider context, including regional differences as well as the original healthcare system of the given country, as mentioned by other authors [3,16] inquiring into the evaluation of technical efficiency of hospitals according to the DEA Model. The results are therefore consistent with the report of the OECD [5], which states that better healthcare leads to a longer life expectancy in the Czech Republic in both men and women, and to a healthier aging process. On the one hand, the Czech Republic is a leader among the Central and Eastern European countries as regards health results; but, on the other hand, the country wrestles with various systemic encumbrances and regional differences which decrease healthcare efficiency.

Based on this research, two problem areas were identified which should be considered by public policy makers on both the national and regional level. This applies to the support in optimizing the structure of human resources in public hospitals. In this regard, it seems advisable to increase the number of nurses while rationalizing the number of other staff, with respect to the special requirements of therapeutic and diagnostic processes in the individual hospitals. In the case of physicians, any interference in their numbers should only be made based on special evaluation processes, because a reduction in the number of physicians is likely to decrease the quality of healthcare [1,6].

## 5. Conclusions

In conclusion, it is worth noting that the process of selecting input and output variables for modeling technical efficiency through the DEA model changes the process of analysis, which acquires objective characteristics and limits subjectivity in measuring these effects. The suitability of the selection of variables is always determined with regard to the objective of the research while respecting the basic rules of construction of DEA models. This also brings a certain limitation to this model, and therefore the results of the analysis cannot be generalized.

## Figures and Tables

**Figure 1 ijerph-18-04711-f001:**
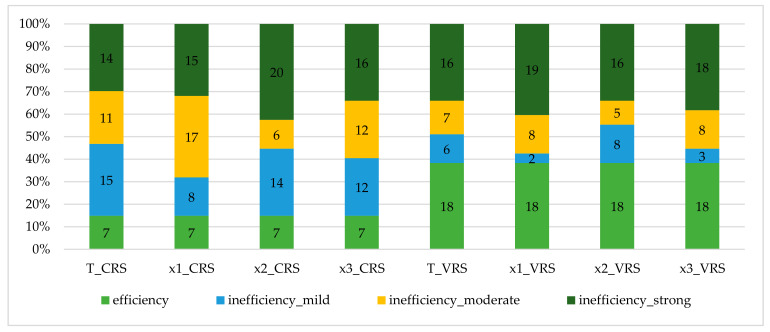
Distribution of the inefficiency rate in the models monitored. CRS: constant returns to scale; VRS: variable returns to scale. T: total efficiency; x1—physicians; x2—nurses; x3—other staff; y1—number of hospitalized patients; y2—number of outpatient examinations; y3—number of surgeries.

**Figure 2 ijerph-18-04711-f002:**
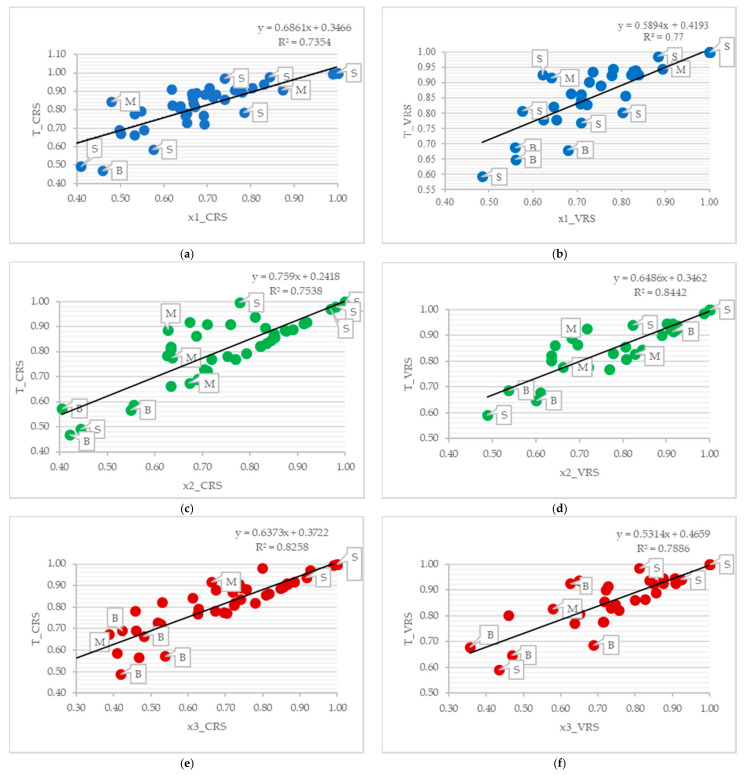
Results within the regression analysis context. (**a**) Regression line T_CRS and x1_CRS. (**b**) Regression line T_VRS and x1_VRS. (**c**) Regression line T_CRS and x2_CRS. (**d**) Regression line T_VRS and x2_VRS. (**e**) Regression line T_CRS and x3_CRS. (**f**) Regression line T_VRS and x3_VRS. Note: S = Small hospitals; M = Medium hospitals; B = Big hospitals.

**Table 1 ijerph-18-04711-t001:** Input and output indicators.

Indicator		Description
Inputs	x1—Number of physicians	Number of professionally competent physicians under professional supervision; professionally competent physicians without professional supervision; physicians with specialized competence.
	x2—Number of nurses	Number of general nurses and midwives.
	x3—Number of other staff	Number of other non-physicians with specialized competence; non-physicians with specialized qualification; non-physicians under professional supervision; other professionals and dentists; technical and economic staff.
Outputs	y1—Number of outpatient treatments	Number of outpatient treatments of patients according to their birth numbers.
	y2—Number of hospitalized patients	Number of hospitalized patients according to the annual results.
	y3—Number of surgeries	Number of surgeries administered.

**Table 2 ijerph-18-04711-t002:** Input and output statistics.

Statistics of Variables	x1	x2	x3	y1	y2	y3
Mean	244.30	653.09	711.28	505,744.20	26,751.95	11,090.66
Median	140.80	410.40	396.60	323,477.70	19,027.30	7311.00
Std. Deviation	232.68	598.27	667.26	431,649.18	21,659.33	10,393.04
Minimum	31.40	110.70	127.60	51,785.70	3870.70	856.00
Maximum	960.60	2663.40	3226.00	1,645,681.30	111,020.00	46,506.70

Note: x1—physicians; x2—nurses; x3—other staff; y1—number of hospitalized patients; y2—number of outpatient examinations; y3—number of surgeries.

**Table 3 ijerph-18-04711-t003:** Hospital efficiency models.

Models	x1	x2	x3	y1	y2	y3
T_CRS/VRS	✓	✓	✓	✓	✓	✓
x1_CRS/VRS	✓			✓	✓	✓
x2_CRS/VRS		✓		✓	✓	✓
x3_CRS/VRS			✓	✓	✓	✓

Note: CRS: constant returns to scale; VRS: variable returns to scale. x1—physicians; x2—nurses; x3—other staff; y1—number of hospitalized patients; y2—number of outpatient examinations; y3—number of surgeries.

**Table 4 ijerph-18-04711-t004:** Average efficiency values.

Models	Mean	Mean in %	Bootstrap			
			Bias	Std. Error	95% Confidence Interval	
					Lower	Upper
T_CRS	0.829	83	−0.001	0.021	0.782	0.868
T_VRS	0.905	91	0.000	0.016	0.871	0.935
x1_CRS	0.703	70	0.000	0.026	0.649	0.755
x1_VRS	0.824	82	0.000	0.024	0.777	0.872
x2_CRS	0.774	77	−0.002	0.024	0.722	0.818
x2_VRS	0.862	86	−0.001	0.022	0.814	0.904
x3_CRS	0.717	72	−0.002	0.030	0.652	0.773
x3_VRS	0.827	83	−0.001	0.028	0.770	0.880

Source: own calculations. Note: CRS: constant returns to scale; VRS: variable returns to scale. T: total efficiency; x1—physicians; x2—nurses; x3—other staff; y1—number of hospitalized patients; y2—number of outpatient examinations; y3—number of surgeries.

**Table 5 ijerph-18-04711-t005:** Correlation matrix.

Models	T_CRS	x1_CRS	x2_CRS	x3_CRS	T_VRS	x1_VRS	x2_VRS	x3_VRS
T_CRS	1							
x1_CRS	0.858 **	1						
x2_CRS	0.868 **	0.743 **	1					
x3_CRS	0.909 **	0.831 **	0.735 **	1				
T_VRS	0.720 **	0.667 **	0.724 **	0.555 **	1			
x1_VRS	0.573 **	0.712 **	0.588 **	0.457 **	0.878 **	1		
x2_VRS	0.597 **	0.567 **	0.766 **	0.415 **	0.919 **	0.827 **	1	
x3_VRS	0.698 **	0.656 **	0.638 **	0.677 **	0.888 **	0.803 **	0.772 **	1

Notes: ** significance α = 0.001. CRS: constant returns to scale; VRS: variable returns to scale. T: total efficiency; x1—physicians; x2—nurses; x3—other staff; y1—number of hospitalized patients; y2—number of outpatient examinations; y3—number of surgeries.

**Table 6 ijerph-18-04711-t006:** Efficiency results according to the hospital size groups.

Models	Small			Medium			Big		
	e = 1	Mean	Min	e = 1	Mean	Min	e = 1	Mean	Min
T_CRS	5	0.858	0.490	1	0.866	0.673	1	0.675	0.469
x1_CRS	5	0.749	0.409	1	0.711	0.479	1	0.540	0.395
x2_CRS	5	0.813	0.444	1	0.776	0.628	1	0.640	0.405
x3_CRS	5	0.761	0.409	1	0.761	0.387	1	0.500	0.294
T_VRS	9	0.909	0.592	5	0.926	0.778	4	0.860	0.648
x1_VRS	9	0.820	0.484	5	0.845	0.623	4	0.806	0.559
x2_VRS	9	0.871	0.488	5	0.867	0.635	4	0.825	0.537
x3_VRS	9	0.837	0.435	5	0.870	0.579	4	0.724	0.357

CRS: constant returns to scale; VRS: variable returns to scale. T: total efficiency; x1—physicians; x2—nurses; x3—other staff; y1—number of hospitalized patients; y2—number of outpatient examinations; y3—number of surgeries.

## Data Availability

The data presented in this study are available on reasonable request from the corresponding author.

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
