# Peer review of "Efficiency of Human Resources in Public Hospitals: An Example from the Czech Republic"

_ijerph, 2021, doi:10.3390/ijerph18094711_

Round 1
Reviewer 1 Report
Thank you for submitting your paper to “IJPERH” with the title of the paper “Efficiency of Human Resources in Public Hospitals: An Example from the Czech Republic”. I have carefully reviewed the manuscript in its current form, the manuscript needs some minor revisions. So, I recommend minor revisions for this paper. I explain some of my reservations in detail below:
- Introduction: The introduction is too long and not written well. This introduction is like an assignment or term paper introduction. The introduction should be much more focused. The research objectives should be much clearer. Perhaps it could be helpful to articulate the research question explicitly. Similarly, authors need to clearly state the paper value-added and better discuss how this work could be worth it for both academics and practitioners. I suggest you explain the answer to mentioned four questions in your revised introduction part. Moreover, I have suggested below mentioned paper it will help you to improve your introduction.
- What are the academic research questions of this study?
- Write the first paragraph about the hospital industry in the Czech Republic and interlink with the human resources. (below-mentioned paper will help you how to write this paragraph).
- I suggest to the authors in the last paragraph of the introduction explain the structure of the paper.
- I also suggest you write your research background like a story
- Rasool SF, Wang M, Tang M, Saeed A, Iqbal J. How Toxic Workplace Environment Effects the Employee Engagement: The Mediating Role of Organizational Support and Employee Wellbeing. International Journal of Environmental Research and Public Health. 2021; 18(5):2294. https://doi.org/10.3390/ijerph18052294.
- Literature Review: Arguments do not flow logically, and ideas are not well connected.
- It is difficult to figure out the research strategy followed. So, I suggest you create a new heading with the title of “Literature review”.
- Under this heading, explain your independent and dependent variables concepts in detail.
- Moreover, explain how these variables are interlinking with each other that affect the hospital performance in the Czech Republic?
- It is also suggested that you explain in detail which theory will support your study and the reflection of this theory will be shown in the overall manuscript (abstract, introduction, literature review, discussion and conclusions).
- Rasool SF, Samma M, Wang M, Zhao Y, Zhang Y. (2019). How Human Resource Management Practices Translate Into Sustainable Organizational Performance: The Mediating Role Of Product, Process And Knowledge Innovation. Psychology Research and Behavior Management, 12, 1009-1025. http://doi.org/10.2147/PRBM.S204662
- Research Methodology: The research methodology of this study needs some explanation. Which research approach the authors used in this study? (Develop sub-heading with the title of Research approach and explain what research approach you used in this study and why you use this research approach specifically in your study).
- Results: Results are fine.
- Discussion and Conclusion: I suggest you split the discussion and conclusions into two parts and explain your discussion with prior studies that will integrate with the literature review.
- Conclusion: The conclusion or policy implication is required to explain in this study. I suggest you explain the to-the-point Conclusion. I also suggested you explain what policies can be improving the efficiency and effectiveness of the small size hospital of the Czech Republic. Furthermore, the conclusion or policy implication must be integrating with the introduction, theory, and your findings.
- References: It is recommended to make use of recent references to support these arguments (ideally, published during the past 5 years).
Good luck with the development of your paper
Author Response
Dear editor,
we very much appreciate all the opinions expressed in the opponents' reviews. Based on their consideration, we have made changes that lead to the improvement of our article.
Ivana Vaňková, Iveta Vrabková

Reviewer 2 Report
- Characterize the performance indicators more fully.
- Characterize the elements of human resources optimization.
- Present the features and the impact on the effectiveness of human resource modeling in public hospitals.
Author Response

(The authors gave the same response as above.)

Reviewer 3 Report
Thank you for the opportunity to review the manuscript entitled “Efficiency of Human Resources in Public Hospitals: An Example from the Czech Republic” (IJERPH-1163246). I think this manuscript is relevant and addresses important issues related to HR and Covid-19. Having said that, I also feel that there are a few minor areas detailed below that need to be addressed by the authors.
- Please edit the words input and output variables with predictor and outcome, respectively (see e.g., Table 1, Table 2, and in-text).
- Please delete the source note beneath each table
- Bootstrapping is not a method; rather, it is a way of computing CIs by random resampling with replacement. Also, please add a brief definition the first time you mention bootstrapping.
- Please add a section in which you discuss the limitations of this paper.
Author Response

(The authors gave the same response as above.)

Reviewer 4 Report
Comments
- The introduction should state the contribution of the paper to the (international) literature.
- The data do not cover all public hospitals. The efficiency and its determinants are most likely heterogeneous across organizations. Does this have implications for the interpretation of the results that are presented in the paper.
- The paper should discuss about the empirical studies (https://doi.org/10.1177/001979391206500203) that have used linked survey and register data to examine the effect of human resources (i.e., job satisfaction) on job performance.
- Does the DEA method have relevant limitations?
- The paper should state more practical policy conclusions that stem from the results.
- The concluding section would benefit from a more balanced discussion of the limitations.
Author Response

(The authors gave the same response as above.)
